# Dual Path Networks

**Yunpeng Chen**[1], **Jianan Li**[1,2], **Huaxin Xiao**[1,3], **Xiaojie Jin**[1], **Shuicheng Yan**[4,1], **Jiashi Feng**[1]
[1]National University of Singapore
[2]Beijing Institute of Technology
[3]National University of Defense Technology
[4]Qihoo 360 AI Institute

## Abstract

In this work, we present a simple, highly efficient and modularized Dual Path Network (DPN) for image classification which presents a new topology of connection paths internally. By revealing the equivalence of the state-of-the-art Residual Network (ResNet) and Densely Convolutional Network (DenseNet) within the HORNN framework, we find that ResNet enables feature re-usage while DenseNet enables new features exploration which are both important for learning good representations. To enjoy the benefits from both path topologies, our proposed Dual Path Network shares common features while maintaining the flexibility to explore new features through dual path architectures. Extensive experiments on three benchmark datasets, ImagNet-1k, Places365 and PASCAL VOC, clearly demonstrate superior performance of the proposed DPN over state-of-the-arts. In particular, on the ImagNet-1k dataset, a shallow DPN surpasses the best ResNeXt-101($64 \times 4$d) with 26% smaller model size, 25% less computational cost and 8% lower memory consumption, and a deeper DPN (DPN-131) further pushes the state-of-the-art single model performance with about 2 times faster training speed. Experiments on the Places365 large-scale scene dataset, PASCAL VOC detection dataset, and PASCAL VOC segmentation dataset also demonstrate its consistently better performance than DenseNet, ResNet and the latest ResNeXt model over various applications.

## 1 Introduction

"Network engineering" is increasingly more important for visual recognition research. In this paper, we aim to develop new path topology of deep architectures to further push the frontier of representation learning. In particular, we focus on analyzing and reforming the skip connection, which has been widely used in designing modern deep neural networks and offers remarkable success in many applications [16, 7, 20, 14, 5]. Skip connection creates a path propagating information from a lower layer directly to a higher layer. During the forward propagation, skip connection enables a very top layer to access information from a distant bottom layer; while for the backward propagation, it facilitates gradient back-propagation to the bottom layer without diminishing magnitude, which effectively alleviates the gradient vanishing problem and eases the optimization.

Deep Residual Network (ResNet) [5] is one of the first works that successfully adopt skip connections, where each mirco-block, *a.k.a.* residual function, is associated with a skip connection, called *residual path*. The residual path element-wisely adds the input features to the output of the same mirco-block, making it a residual unit. Depending on the inner structure design of the mirco-block, the residual network has developed into a family of various architectures, including WRN [22], Inception-resnet [20], and ResNeXt [21].

More recently, Huang et al. [8] proposed a different network architecture that achieves comparable accuracy with deep ResNet [5], named Dense Convolutional Network (DenseNet). Different from residual networks which add the input features to the output features through the residual path, the

DenseNet uses a *densely connected path* to concatenate the input features with the output features, enabling each micro-block to receive raw information from all previous micro-blocks. Similar with residual network family, DenseNet can be categorized to the densely connected network family. Although the width of the densely connected path increases linearly as it goes deeper, causing the number of parameters to grow quadratically, DenseNet provides higher parameter efficiency compared with the ResNet [5].

In this work, we aim to study the advantages and limitations of both topologies and further enrich the path design by proposing a dual path architecture. In particular, we first provide a new understanding of the densely connected networks from the lens of a higher order recurrent neural network (HORNN) [19], and explore the relations between densely connected networks and residual networks. More specifically, we bridge the densely connected networks with the HORNNs, showing that the densely connected networks are HORNNs when the weights are shared across steps. Inspired by [12] which demonstrates the relations between the residual networks and RNNs, we prove that the residual networks are densely connected networks when connections are shared across layers. With this unified view on the state-of-the-art deep architecture, we find that the deep residual networks implicitly reuse the features through the residual path, while densely connected networks keep exploring new features through the densely connected path.

Based on this new view, we propose a novel dual path architecture, called the Dual Path Network (DPN). This new architecture inherits both advantages of residual and densely connected paths, enabling effective feature re-usage and re-exploitation. The proposed DPN also enjoys higher parameter efficiency, lower computational cost and lower memory consumption, and being friendly for optimization compared with the state-of-the-art classification networks. Experimental results validate the outstanding high accuracy of DPN compared with other well-established baselines for image classification on both ImageNet-1k dataset and Places365-Standard dataset. Additional experiments on object detection task and semantic segmentation task also demonstrate that the proposed dual path architecture can be broadly applied for various tasks and consistently achieve the best performance.

## 2 Related work

Designing an advanced neural network architecture is one of the most challenging but effective ways for improving the image classification performance, which can also directly benefit a variety of other tasks. AlexNet [10] and VGG [18] are two most important works that show the power of deep convolutional neural networks. They demonstrate that building deeper networks with tiny convolutional kernels is a promising way to increase the learning capacity of the neural network. Residual Network was first proposed by He et al. [5], which greatly alleviates the optimization difficulty and further pushes the depth of deep neural networks to hundreds of layers by using skipping connections. Since then, different kinds of residual networks arose, concentrating on either building a more efficient micro-block inner structure [3, 21] or exploring how to use residual connections [9]. Recently, Huang et al. [8] proposed a different network, called Dense Convolutional Networks, where skip connections are used to concatenate the input to the output instead of adding. However, the width of the densely connected path linearly increases as the depth rises, causing the number of parameters to grow quadratically and costing a large amount of GPU memory compared with the residual networks if the implementation is not specifically optimized. This limits the building of a deeper and wider densenet that may further improve the accuracy.

Besides designing new architectures, researchers also try to re-explore the existing state-of-the-art architectures. In [6], the authors showed the importance of the residual path on alleviating the optimization difficulty. In [12], the residual networks are bridged with recurrent neural networks (RNNs), which helps people better understand the deep residual network from the perspective of RNNs. In [3], several different residual functions are unified, trying to provide a better understanding of designing a better mirco structure with higher learning capacity. But still, for the densely connected networks, in addition to several intuitive explanations on better feature reusage and efficient gradient flow introduced, there have been few works that are able to provide a really deeper understanding.

In this work, we provide a deeper understanding of the densely connected network, from the lens of Higher Order RNN, and explain how the residual networks are in indeed a special case of densely connected network. Based on these analysis, we then propose a novel Dual Path Network architecture that not only achieves higher accuracy, but also enjoys high parameter and computational efficiency.

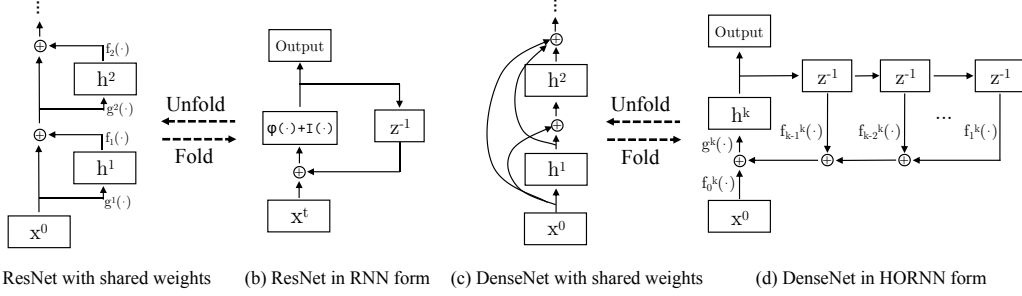

(a) ResNet with shared weights  (b) ResNet in RNN form  (c) DenseNet with shared weights  (d) DenseNet in HORNN form

Figure 1: The topological relations of different types of neural networks. (a) and (b) show relations between residual networks and RNN, as stated in [12]; (c) and (d) show relations between densely connected networks and higher order recurrent neural network (HORNN), which is explained in this paper. The symbol "$z^{-1}$" denotes a time-delay unit; "$\oplus$" denotes the element-wise summation; "$I(\cdot)$" denotes an identity mapping function.

## 3  Revisiting ResNet, DenseNet and Higher Order RNN

In this section, we first bridge the densely connected network [8] with higher order recurrent neural networks [19] to provide a new understanding of the densely connected network. We prove that residual networks [5, 6, 22, 21, 3], essentially belong to the family of densely connected networks except their connections are shared across steps. Then, we present analysis on strengths and weaknesses of each topology architecture, which motivates us to develop the dual path network architecture.

For exploring the above relation, we provide a new view on the densely connected networks from the lens of Higher Order RNN, explain their relations and then specialize the analysis to residual networks. Throughout the paper, we formulate the HORNN in a more generalized form. We use $h^t$ to denote the hidden state of the recurrent neural network at the $t$-th step and use $k$ as the index of the current step. Let $x^t$ denotes the input at $t$-th step, $h^0 = x^0$. For each step, $f_t^k(\cdot)$ refers to the feature extracting function which takes the hidden state as input and outputs the extracted information. The $g^k(\cdot)$ denotes a transformation function that transforms the gathered information to current hidden state:

$$h^k = g^k \left[ \sum_{t=0}^{k-1} f_t^k(h^t) \right].$$ (1)

Eqn. (1) encapsulates the update rule of various network architectures in a generalized way. For HORNNs, weights are shared across steps, *i.e.* $\forall t, k, f_{k-t}^k(\cdot) \equiv f_t(\cdot)$ and $\forall k, g^k(\cdot) \equiv g(\cdot)$. For the densely connected networks, each step (micro-block) has its own parameter, which means $f_t^k(\cdot)$ and $g^k(\cdot)$ are not shared. Such observation shows that the densely connected path of DenseNet is essentially a higher order path which is able to extract new information from previous states. Figure 1(c)(d) graphically shows the relations of densely connected networks and higher order recurrent networks.

We then explain that the residual networks are special cases of densely connected networks if taking $\forall t, k, f_t^k(\cdot) \equiv f_t(\cdot)$. Here, for succinctness we introduce $r^k$ to denote the intermediate results and let $r^0 = 0$. Then Eqn. (1) can be rewritten as

$$r^k \triangleq \sum_{t=1}^{k-1} f_t(h^t) = r^{k-1} + f_{k-1}(h^{k-1}),$$ (2)

$$h^k = g^k\left(r^k\right).$$ (3)

Thus, by substituting Eqn. (3) into Eqn. (2), Eqn. (2) can be simplified as

$$r^k = r^{k-1} + f_{k-1}(h^{k-1}) = r^{k-1} + f_{k-1}(g^{k-1}\left(r^{k-1}\right)) = r^{k-1} + \phi^{k-1}(r^{k-1}),$$ (4)

where $\phi^k(\cdot) = f_k(g^k(\cdot))$. Obviously, Eqn. (4) has the same form as the residual network and the recurrent neural network. Specifically, when $\forall k, \phi^k(\cdot) \equiv \phi(\cdot)$, Eqn. (4) degenerates to an RNN; when none of $\phi^k(\cdot)$ is shared and $x^k = 0, k > 1$, Eqn. (4) produces a residual network. Figure 1(a)(b)

graphically shows the relation. Besides, recall that Eqn. (4) is derived under the condition when $\forall t, k, f_t^k(\cdot) \equiv f_t(\cdot)$ from Eqn. (1) and the densely connected networks are in forms of Eqn. (1), meaning that the residual network family essentially belongs to the densely connected network family. Figure 2(a–c) give an example and demonstrate such equivalence, where $f_t(\cdot)$ corresponds to the first $1 \times 1$ convolutional layer and the $g^k(\cdot)$ corresponds to the other layers within a micro-block in Figure 2(b).

From the above analysis, we observe: 1) both residual networks and densely connected networks can be seen as a HORNN when $f_t^k(\cdot)$ and $g^k(\cdot)$ are shared for all $k$; 2) a residual network is a densely connected network if $\forall t, k, f_t^k(\cdot) \equiv f_t(\cdot)$. By sharing the $f_t^k(\cdot)$ across all steps, $g^k(\cdot)$ receives the same feature from a given output state, which encourages the feature reusage and thus reduces the feature redundancy. However, such an information sharing strategy makes it difficult for residual networks to explore new features. Comparatively, the densely connected networks are able to explore new information from previous outputs since the $f_t^k(\cdot)$ is not shared across steps. However, different $f_t^k(\cdot)$ may extract the same type of features multiple times, leading to high redundancy.

In the following section, we present the dual path networks which can overcome both inherent limitations of these two state-of-the-art network architectures. Their relations with HORNN also imply that our proposed architecture can be used for improving HORNN, which we leave for future works.

## 4  Dual Path Networks

Above we explain the relations between residual networks and densely connected networks, showing that the residual path implicitly reuses features, but it is not good at exploring new features. In contrast the densely connected network keeps exploring new features but suffers from higher redundancy.

In this section, we describe the details of our proposed novel dual path architecture, *i.e.* the Dual Path Network (DPN). In the following, we first introduce and formulate the dual path architecture, and then present the network structure in details with complexity analysis.

### 4.1  Dual Path Architecture

Sec. 3 discusses the advantage and limitations of both residual networks and densely connected networks. Based on the analysis, we propose a simple dual path architecture which shares the $f_t^k(\cdot)$ across all blocks to enjoy the benefits of reusing common features with low redundancy, while still remaining a densely connected path to give the network more flexibility in learning new features. We formulate such a dual path architecture as follows:

$$x^k \triangleq \sum_{t=1}^{k-1} f_t^k(h^t), \tag{5}$$

$$y^k \triangleq \sum_{t=1}^{k-1} v_t(h^t) = y^{k-1} + \phi^{k-1}(y^{k-1}), \tag{6}$$

$$r^k \triangleq x^k + y^k, \tag{7}$$

$$h^k = g^k\left(r^k\right), \tag{8}$$

where $x^k$ and $y^k$ denote the extracted information at $k$-th step from individual path, $v_t(\cdot)$ is a feature learning function as $f_t^k(\cdot)$. Eqn. (5) refers to the densely connected path that enables exploring new features, Eqn. (6) refers to the residual path that enables common features re-usage, and Eqn. (7) defines the dual path that integrates them and feeds them to the last transformation function in Eqn. (8). The final transformation function $g^k(\cdot)$ generates current state, which is used for making next mapping or prediction. Figure 2(d)(e) show an example of the dual path architecture that is being used in our experiments.

More generally, the proposed DPN is a family of convolutional neural networks which contains a residual alike path and a densely connected alike path, as explained later. Similar to these networks, one can customize the micro-block function of DPN for task-specific usage or for further overall performance boosting.

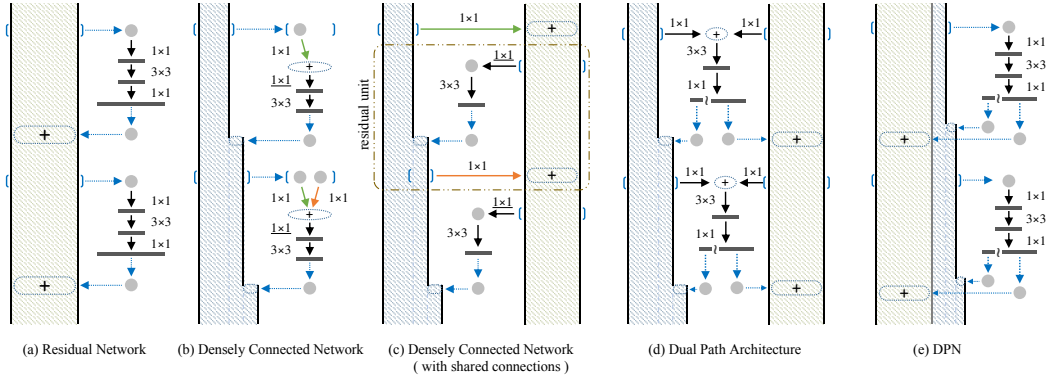

| (a) Residual Network | (b) Densely Connected Network | (c) Densely Connected Network ( with shared connections ) | (d) Dual Path Architecture | (e) DPN |

Figure 2: Architecture comparison of different networks. (a) The residual network. (b) The densely connected network, where each layer can access the outputs of all previous micro-blocks. Here, a $1 \times 1$ convolutional layer (underlined) is added for consistency with the micro-block design in (a). (c) By sharing the first $1 \times 1$ connection of the same output across micro-blocks in (b), the densely connected network degenerates to a residual network. The dotted rectangular in (c) highlights the residual unit. (d) The proposed dual path architecture, DPN. (e) An equivalent form of (d) from the perspective of implementation, where the symbol "⅃" denotes a split operation, and "+" denotes element-wise addition.

## 4.2 Dual Path Networks

The proposed network is built by stacking multiple modualized mirco-blocks as shown in Figure 2. In this work, the structure of each micro-block is designed with a bottleneck style [5] which starts with a $1 \times 1$ convolutional layer followed by a $3 \times 3$ convolutional layer, and ends with a $1 \times 1$ convolutional layer. The output of the last $1 \times 1$ convolutional layer is split into two parts: the first part is element-wisely added to the residual path, and the second part is concatenated with the densly connected path. To enhance the leaning capacity of each micro-block, we use the grouped convolution layer in the second layer as the ResNeXt [21].

Considering that the residual networks are more wildly used than the densely connected networks in practice, we choose the residual network as the backbone and add a thin densely connected path to build the dual path network. Such design also helps slow the width increment of the densely connected path and the cost of GPU memory. Table 1 shows the detailed architecture settings. In the table, $G$ refers to the number of groups, and $k$ refers to the channels increment for the densely connected path. For the new proposed DPNs, we use $(+k)$ to indicate the width increment of the densely connected path. The overall design of DPN inherits backbone architecture of the vanilla ResNet / ResNeXt, making it very easy to implement and apply to other tasks. One can simply implement a DPN by adding one more "slice layer" and "concat layer" upon existing residual networks. Under a well optimized deep learning platform, none of these newly added operations requires extra computational cost or extra memory consumption, making the DPNs highly efficient.

In order to demonstrate the appealing effectiveness of the dual path architecture, we intentionally design a set of DPNs with a considerably smaller model size and less FLOPs compared with the sate-of-the-art ResNeXts [21], as shown in Table 1. Due to limited computational resources, we set these hyper-parameters based on our previous experience instead of grid search experiments.

**Model complexity** We measure the model complexity by counting the total number of learnable parameters within each neural network. Table 1 shows the results for different models. The DPN-92 costs about $15\%$ fewer parameters than ResNeXt-101 ($32 \times 4$d), while the DPN-98 costs about $26\%$ fewer parameters than ResNeXt-101 ($64 \times 4$d).

**Computational complexity** We measure the computational cost of each deep neural network using the floating-point operations (FLOPs) with input size of $224 \times 224$, in the number of multiply-adds following [21]. Table 1 shows the theoretical computational cost. Though the actual time cost might be influenced by other factors, *e.g.* GPU bandwidth and coding quality, the computational cost shows the speed upper bound. As can be see from the results, DPN-92 consumes about $19\%$ less FLOPs than ResNeXt-101($32 \times 4$d), and the DPN-98 consumes about $25\%$ less FLOPs than ResNeXt-101($64 \times 4$d).

Table 1: Architecture and complexity comparison of our proposed Dual Path Networks (DPNs) and other state-of-the-art networks. We compare DPNs with two baseline methods: DenseNet [5] and ResNeXt [21]. The symbol $(+k)$ denotes the width increment on the densely connected path.

| stage | output | DenseNet-161 (k=48) | ResNeXt-101 (32×4d) | ResNeXt-101 (64×4d) | DPN-92 (32×3d) | DPN-98 (40×4d) |
|---|---|---|---|---|---|---|
| conv1 | 112x112 | 7 × 7, 96, stride 2 | 7 × 7, 64, stride 2 | 7 × 7, 64, stride 2 | 7 × 7, 64, stride 2 | 7 × 7, 96, stride 2 |
| conv2 | 56×56 | 3 × 3 max pool, stride 2 <br> $\begin{bmatrix} 1\times1,\,192 \\ 3\times3,\,48 \end{bmatrix} \times 6$ | 3 × 3 max pool, stride 2 <br> $\begin{bmatrix} 1\times1,\,128 \\ 3\times3,\,128,\,G{=}32 \\ 1\times1,\,256 \end{bmatrix} \times 3$ | 3 × 3 max pool, stride 2 <br> $\begin{bmatrix} 1\times1,\,256 \\ 3\times3,\,256,\,G{=}64 \\ 1\times1,\,256 \end{bmatrix} \times 3$ | 3 × 3 max pool, stride 2 <br> $\begin{bmatrix} 1\times1,\,96 \\ 3\times3,\,96,\,G{=}32 \\ 1\times1,\,256\ (+16) \end{bmatrix} \times 3$ | 3 × 3 max pool, stride 2 <br> $\begin{bmatrix} 1\times1,\,160 \\ 3\times3,\,160,\,G{=}40 \\ 1\times1,\,256\ (+16) \end{bmatrix} \times 3$ |
| conv3 | 28×28 | $\begin{bmatrix} 1\times1,\,192 \\ 3\times3,\,48 \end{bmatrix} \times 12$ | $\begin{bmatrix} 1\times1,\,256 \\ 3\times3,\,256,\,G{=}32 \\ 1\times1,\,512 \end{bmatrix} \times 4$ | $\begin{bmatrix} 1\times1,\,512 \\ 3\times3,\,512,\,G{=}64 \\ 1\times1,\,512 \end{bmatrix} \times 4$ | $\begin{bmatrix} 1\times1,\,192 \\ 3\times3,\,192,\,G{=}32 \\ 1\times1,\,512\ (+32) \end{bmatrix} \times 4$ | $\begin{bmatrix} 1\times1,\,320 \\ 3\times3,\,320,\,G{=}40 \\ 1\times1,\,512\ (+32) \end{bmatrix} \times 6$ |
| conv4 | 14×14 | $\begin{bmatrix} 1\times1,\,192 \\ 3\times3,\,48 \end{bmatrix} \times 36$ | $\begin{bmatrix} 1\times1,\,512 \\ 3\times3,\,512,\,G{=}32 \\ 1\times1,\,1024 \end{bmatrix} \times 23$ | $\begin{bmatrix} 1\times1,\,1024 \\ 3\times3,\,1024,\,G{=}64 \\ 1\times1,\,1024 \end{bmatrix} \times 23$ | $\begin{bmatrix} 1\times1,\,384 \\ 3\times3,\,384,\,G{=}32 \\ 1\times1,\,1024\ (+24) \end{bmatrix} \times 20$ | $\begin{bmatrix} 1\times1,\,640 \\ 3\times3,\,640,\,G{=}40 \\ 1\times1,\,1024\ (+32) \end{bmatrix} \times 20$ |
| conv5 | 7×7 | $\begin{bmatrix} 1\times1,\,192 \\ 3\times3,\,48 \end{bmatrix} \times 24$ | $\begin{bmatrix} 1\times1,\,1024 \\ 3\times3,\,1024,\,G{=}32 \\ 1\times1,\,2048 \end{bmatrix} \times 3$ | $\begin{bmatrix} 1\times1,\,2048 \\ 3\times3,\,2048,\,G{=}64 \\ 1\times1,\,2048 \end{bmatrix} \times 3$ | $\begin{bmatrix} 1\times1,\,768 \\ 3\times3,\,768,\,G{=}32 \\ 1\times1,\,2048\ (+128) \end{bmatrix} \times 3$ | $\begin{bmatrix} 1\times1,\,1280 \\ 3\times3,\,1280,\,G{=}40 \\ 1\times1,\,2048\ (+128) \end{bmatrix} \times 3$ |
| | 1×1 | global average pool 1000-d fc, softmax | global average pool 1000-d fc, softmax | global average pool 1000-d fc, softmax | global average pool 1000-d fc, softmax | global average pool 1000-d fc, softmax |
| # params | | $28.9 \times 10^6$ | $44.3 \times 10^6$ | $83.7 \times 10^6$ | $37.8 \times 10^6$ | $61.7 \times 10^6$ |
| FLOPs | | $7.7 \times 10^9$ | $8.0 \times 10^9$ | $15.5 \times 10^9$ | $6.5 \times 10^9$ | $11.7 \times 10^9$ |

# 5 Experiments

Extensive experiments are conducted for evaluating the proposed Dual Path Networks. Specifically, we evaluate the proposed architecture on three tasks: image classification, object detection and semantic segmentation, using three standard benchmark datasets: the ImageNet-1k dataset, Places365-Standard dataset and the PASCAL VOC datasets.

Key properties of the proposed DPNs are studied on the ImageNet-1k object classification dataset [17] and further verified on the Places365-Standard scene understanding dataset [24]. To verify whether the proposed DPNs can benefit other tasks besides image classification, we further conduct experiments on the PASCAL VOC dataset [4] to evaluate its performance in object detection and semantic segmentation.

## 5.1 Experiments on image classification task

We implement the DPNs using MXNet [2] on a cluster with 40 K80 graphic cards. Following [3], we adopt standard data augmentation methods and train the networks using SGD with a mini-batch size of 32 for each GPU. For the deepest network, *i.e.* DPN-131[1], the mini-batch size is limited to 24 because of the 12GB GPU memory constraint. The learning rate starts from $\sqrt{0.1}$ for DPN-92 and DPN-131, and from $0.4$ for DPN-98. It drops in a "steps" manner by a factor of $0.1$. Following [5], batch normalization layers are refined after training.

### 5.1.1 ImageNet-1k dataset

Firstly, we compare the image classification performance of DPNs with current state-of-the-art models. As can be seen from the first block in Table 2, a shallow DPN with only the depth of 92 reduces the top-1 error rate by an absolute value of $0.5\%$ compared with the ResNeXt-101($32 \times 4d$) and an absolute value of $1.5\%$ compared with the DenseNet-161 yet provides with considerably less FLOPs. In the second block of Table 2, a deeper DPN (DPN-98) surpasses the best residual network – ResNeXt-101 ($64 \times 4d$), and still enjoys $25\%$ less FLOPs and a much smaller model size (236 MB v.s. 320 MB). In order to further push the state-of-the-art accuracy, we slightly increase the depth of the DPN to 131 (DPN-131). The results are shown in the last block in Table 2. Again, the DPN shows superior accuracy over the best single model – Very Deep PolyNet [23], with a much smaller model size (304 MB v.s. 365 MB). Note that the Very Deep PolyNet adopts numerous tricks, *e.g.* initialization by insertion, residual scaling, stochastic paths, to assist the training process. In contrast, our proposed DPN-131 is simple and does not involve these tricks, DPN-131 can be trained using a standard training strategy as shallow DPNs. More importantly, the actual training speed of DPN-131 is about 2 times faster than the Very Deep PolyNet, as discussed in the following paragraph.

Table 2: Comparison with state-of-the-art CNNs on ImageNet-1k dataset. Single crop validation error rate (%) on validation set. *: Performance reported by [21], †: With Mean-Max Pooling (see supplementary material).

| Method | Model Size | GFLOPs | x224 top-1 | x224 top-5 | x320 / x299 top-1 | x320 / x299 top-5 |
|---|---|---|---|---|---|---|
| DenseNet-161(k=48) [8] | 111 MB | 7.7 | 22.2 | – | – | – |
| ResNet-101* [5] | 170 MB | 7.8 | 22.0 | 6.0 | – | – |
| ResNeXt-101 (32 × 4d) [21] | 170 MB | 8.0 | 21.2 | 5.6 | – | – |
| **DPN-92 (32 × 3d)** | **145 MB** | **6.5** | **20.7** | **5.4** | **19.3** | **4.7** |
| ResNet-200 [6] | 247 MB | 15.0 | 21.7 | 5.8 | 20.1 | 4.8 |
| Inception-resnet-v2 [20] | 227 MB | – | – | – | 19.9 | 4.9 |
| ResNeXt-101 (64 × 4d) [21] | 320 MB | 15.5 | 20.4 | 5.3 | 19.1 | 4.4 |
| **DPN-98 (40 × 4d)** | **236 MB** | **11.7** | **20.2** | **5.2** | **18.9** | **4.4** |
| Very deep Inception-resnet-v2 [23] | 531 MB | – | – | – | 19.10 | 4.48 |
| Very Deep PolyNet [23] | 365 MB | – | – | – | 18.71 | 4.25 |
| DPN-131 (40 × 4d) | 304 MB | 16.0 | 19.93 | 5.12 | 18.62 | 4.23 |
| **DPN-131 (40 × 4d) †** | **304 MB** | **16.0** | **19.93** | **5.12** | **18.55** | **4.16** |

Table 3: Comparison with state-of-the-art CNNs on Places365-Standard dataset. 10 crops validation accuracy rate (%) on validation set.

| Method | Model Size | top-1 acc. | top-5 acc. |
|---|---|---|---|
| AlexNet [24] | 223 MB | 53.17 | 82.89 |
| GoogleLeNet [24] | 44 MB | 53.63 | 83.88 |
| VGG-16 [24] | 518 MB | 55.24 | 84.91 |
| ResNet-152 [24] | 226 MB | 54.74 | 85.08 |
| ResNeXt-101 [3] | 165 MB | 56.21 | 86.25 |
| CRU-Net-116 [3] | 163 MB | 56.60 | 86.55 |
| **DPN-92 (32 × 3d)** | **138 MB** | **56.84** | **86.69** |

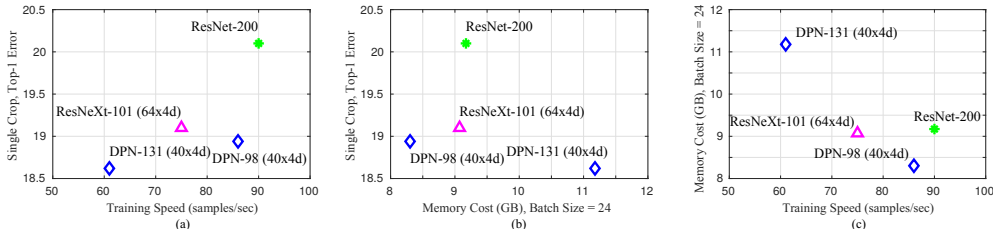

Figure 3: Comparison of total actual cost between different models during training. Evaluations are conducted on a single Node with 4 K80 graphic card with all training samples cached into memory. (For the comparison of *Training Speed*, we push the mini-batch size to its maximum value given a 12GB GPU memory to test the fastest possible training speed of each model.)

Secondly, we compare the training cost between the best performing models. Here, we focus on evaluating two key properties – the actual GPU memory cost and the actual training speed. Figure 3 shows the results. As can be seen from Figure 3(a)(b), the DPN-98 is 15% faster and uses 9% less memory than the best performing ResNeXt with a considerably lower testing error rate. Note that theoretically the computational cost of DPN-98 shown in Table 2 is 25% less than the best performing ResNeXt, indicating there is still room for code optimization. Figure 3(c) presents the same result in a more clear way. The deeper DPN-131 only costs about 19% more training time compared with the best performing ResNeXt, but achieves the state-of-the-art single model performance. The training speed of the previous state-of-the-art single model, *i.e.* Very Deep PolyNet (537 layers) [23], is about 31 samples per second based on our implementation using MXNet, showing that DPN-131 runs about 2 times faster than the Very Deep PolyNet during training.

### 5.1.2 Place365-Standard dataset

In this experiment, we further evaluate the accuracy of the proposed DPN on the scene classification task using the Places365-Standard dataset. The Places365-Standard dataset is a high-resolution scene understanding dataset with more than 1.8 million images of 365 scene categories. Different from object images, scene images do not have very clear discriminative patterns and require a higher level context reasoning ability.

Table 3 shows the results of different models on this dataset. To make a fair comparison, we perform the DPN-92 on this dataset instead of using deeper DPNs. As can be seen from the results, DPN achieves the best validation accuracy compared with other methods. The DPN-92 requires much less parameters (138 MB v.s. 163 MB), which again demonstrates its high parameter efficiency and high generalization ability.

## 5.2 Experiments on the object detection task

We further evaluate the proposed Dual Path Network on the object detection task. Experiments are performed on the PASCAL VOC 2007 datasets [4]. We train the models on the union set of VOC 2007 *trainval* and VOC 2012 *trainval* following [16], and evaluate them on VOC 2007 *test* set. We use standard evaluation metrics Average Precision (AP) and mean of AP (mAP) following the PASCAL challenge protocols for evaluation.

Table 4: Object detection results on PASCAL VOC 2007 *test* set. The performance is measured by mean of Average Precision (mAP, in %).

| Method | mAP | areo | bike | bird | boat | bottle | bus | car | cat | chair | cow | table | dog | horse | mbk | prsn | plant | sheep | sofa | train | tv |
|---|---|---|---|---|---|---|---|---|---|---|---|---|---|---|---|---|---|---|---|---|---|
| DenseNet-161 (k=48) | 79.9 | 80.4 | 85.9 | 81.2 | 72.8 | 68.0 | 87.1 | 88.0 | 88.8 | 64.0 | 83.3 | 75.4 | 87.5 | 87.6 | 81.3 | 84.2 | 54.6 | 83.2 | 80.2 | 87.4 | 77.2 |
| ResNet-101 [16] | 76.4 | 79.8 | 80.7 | 76.2 | 68.3 | 55.9 | 85.1 | 85.3 | **89.8** | 56.7 | **87.8** | 69.4 | 88.3 | **88.9** | 80.9 | 78.4 | 41.7 | 78.6 | 79.8 | 85.3 | 72.0 |
| ResNeXt-101 (32 × 4d) | 80.1 | 80.2 | 86.5 | 79.4 | 72.5 | 67.3 | 86.9 | 88.6 | 88.9 | 64.9 | 85.0 | 76.2 | 87.3 | 87.8 | 81.8 | 84.1 | 55.5 | 84.0 | 79.7 | 87.9 | 77.0 |
| DPN-92 (32 × 3d) | **82.5** | **84.4** | **88.5** | **84.6** | **76.5** | **70.7** | **87.9** | **88.8** | 89.4 | **69.7** | 87.0 | **76.7** | **89.5** | 88.7 | **86.0** | **86.1** | **58.4** | **85.0** | **80.4** | **88.2** | **83.1** |

Table 5: Semantic segmentation results on PASCAL VOC 2012 *test* set. The performance is measured by mean Intersection over Union (mIoU, in %).

| Method | mIoU | bkg | areo | bike | bird | boat | bottle | bus | car | cat | chair | cow | table | dog | horse | mbk | prsn | plant | sheep | sofa | train | tv |
|---|---|---|---|---|---|---|---|---|---|---|---|---|---|---|---|---|---|---|---|---|---|---|
| DenseNet-161 (k=48) | 68.7 | 92.1 | 77.3 | 37.1 | 83.6 | 54.9 | 70.0 | 85.8 | 82.5 | 85.9 | 26.1 | 73.0 | 55.1 | 80.2 | 74.0 | 79.1 | 78.2 | 51.5 | 80.0 | 42.2 | 75.1 | 58.6 |
| ResNet-101 | 73.1 | 93.1 | 86.9 | 39.9 | **87.6** | 59.6 | 74.4 | 90.1 | 84.7 | 87.7 | 30.0 | 81.8 | 56.2 | 82.7 | 82.7 | 80.1 | 81.1 | 52.4 | 86.2 | 52.5 | 81.3 | 63.6 |
| ResNeXt-101 (32 × 4d) | 73.6 | 93.1 | 84.9 | 36.2 | 80.3 | **65.0** | **74.7** | 90.6 | 83.9 | 88.7 | **31.1** | 86.3 | **62.4** | **84.7** | 86.1 | 81.2 | 80.1 | 54.0 | **87.4** | 54.0 | 76.3 | 64.2 |
| DPN-92 (32 × 3d) | **74.8** | **93.7** | **88.3** | **40.3** | 82.7 | 64.5 | 72.0 | **90.9** | **85.0** | **88.8** | **31.1** | **87.7** | 59.8 | 83.9 | **86.8** | **85.1** | **82.8** | **60.8** | 85.3 | **54.1** | **82.6** | **64.6** |

We perform all experiments based on the ResNet-based Faster R-CNN framework, following [5] and make comparisons by replacing the ResNet, while keeping other parts unchanged. Since our goal is to evaluate DPN, rather than further push the state-of-the-art accuracy on this dataset, we adopt the shallowest DPN-92 and baseline networks at roughly the same complexity level. Table 4 provides the detection performance comparisons of the proposed DPN with several current state-of-the-art models. It can be observed that the DPN obtains the mAP of 82.5%, which makes large improvements, *i.e.* 6.1% compared with ResNet-101 [16] and 2.4% compared with ResNeXt-101 (32 × 4d). The better results shown in this experiment demonstrate that the Dual Path Network is also capable of learning better feature representations for detecting objects and benefiting the object detection task.

## 5.3 Experiments on the semantic segmentation task

In this experiment, we evaluate the Dual Path Network for dense prediction, *i.e.* semantic segmentation, where the training target is to predict the semantic label for each pixel in the input image. We conduct experiments on the PASCAL VOC 2012 segmentation benchmark dataset [4] and use the DeepLab-ASPP-L [1] as the segmentation framework. For each compared method in Table 5, we replace the $3 \times 3$ convolutional layers in conv4 and conv5 of Table 1 with atrous convolution [1] and plug in a head of Atrous Spatial Pyramid Pooling (ASPP) [1] in the final feature maps of conv5. We adopt the same training strategy for all networks following [1] for fair comparison.

Table 5 shows the results of different convolutional neural networks. It can be observed that the proposed DPN-92 has the highest overall mIoU accuracy. Compared with the ResNet-101 which has a larger model size and higher computational cost, the proposed DPN-92 further improves the IoU for most categories and improves the overall mIoU by an absolute value 1.7%. Considering the ResNeXt-101 (32 × 4d) only improves the overall mIoU by an absolute value 0.5% compared with the ResNet-101, the proposed DPN-92 gains more than 3 times improvement compared with the ResNeXt-101 (32 × 4d). The better results once again demonstrate the proposed Dual Path Network is capable of learning better feature representation for dense prediction.

## 6 Conclusion

In this paper, we revisited the densely connected networks, bridged the densely connected networks with Higher Order RNNs and proved the residual networks are essentially densely connected networks with shared connections. Based on this new explanation, we proposed a dual path architecture that enjoys benefits from both sides. The novel network, DPN, is then developed based on this dual path architecture. Experiments on the image classification task demonstrate that the DPN enjoys high accuracy, small model size, low computational cost and low GPU memory consumption, thus is extremely useful for not only research but also real-word application. Experiments on the object detection task and semantic segmentation tasks show that the proposed DPN can also benefit other tasks by simply replacing the base network.

**Acknowledgments**
The work of Jiashi Feng was partially supported by National University of Singapore startup grant R-263-000-C08-133, Ministry of Education of Singapore AcRF Tier One grant R-263-000-C21-112 and NUS IDS grant R-263-000-C67-646.

## Footnotes

[1]The DPN-131 has 128 channels at conv1, 4 blocks at conv2, 8 blocks at conv3, 28 blocks at conv4 and 3 blocks at conv5, which has #params=$79.5 \times 10^6$ and FLOPs=$16.0 \times 10^9$.

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
