[Supplementary Material]

# Dual Path Networks
# – Supplementary Material

**Yunpeng Chen[1], Jianan Li[1,2], Huaxin Xiao[1,3], Xiaojie Jin[1], Shuicheng Yan[4,1], Jiashi Feng[1]**
[1]National University of Singapore
[2]Beijing Institute of Technology
[3]National University of Defense Technology
[4]Qihoo 360 AI Institute

## 1 Testing with Mean-Max Pooling

Here, we introduce a new testing technique by using Mean-Max Pooling which can further improve the performance of a well trained CNN in the testing phase without any noticeable computational overhead. This testing technique is very effective for testing images with size larger than training crops. The idea is to first convert a trained CNN model into a convolutional network [2] and then insert the following Mean-Max Pooling layer (*a.k.a.* Max-Avg Pooling [1]), *i.e.* 0.5 * (global average pooling + global max pooling), just before the final softmax layer.

Table 1: Comparison with different testing techniques on ImageNet-1k dataset. Single crop validation error rate (%) on validation set.

| Method | Model Size | GFLOPs | w/o Mean-Max Pooling | | w/ Mean-Max Pooling | |
|---|---|---|---|---|---|---|
| | | | top-1 | top-5 | top-1 | top-5 |
| DPN-92 ($32 \times 3d$) | 145 MB | 6.5 | 19.34 | 4.66 | **19.04** | **4.53** |
| DPN-98 ($40 \times 4d$) | 236 MB | 11.7 | 18.94 | 4.44 | **18.72** | **4.40** |
| DPN-131 ($40 \times 4d$) | 304 MB | 16.0 | 18.62 | 4.23 | **18.55** | **4.16** |

Comparisons between the models with and without Mean-Max Pooling are shown in Table 1. As can be seen from the results, the simple Mean-Max Pooling testing strategy successfully improves the testing accuracy for all models.