[Reviews · NeurIPS 2017]

Reviewer 1



The paper proposes a new CNN architecture named Dual Path Network (DPN), which carefully blends elements from ResNets and DenseNets. The paper studies ResNets, DenseNets, and the newly proposed DPNs from the perspective of RNNs. It then proposes a careful hybrid design which combines the benefits of both ResNet and DenseNet models. The paper comes with a very strong experimental evaluation. The authors compare the proposed DPN with other state-of-art architectures (ResNet, Inception-ResNet, ResNeXt) and show favorable results for comparable or computational and memory cost, not only on ImageNet classification but also on other visual recognition tasks (object detection, semantic segmentation, etc). Given the importance of high-performing network backbones in computer vision tasks and the strong results presented in this paper, I find that this paper should clearly be accepted for publication at this conference.

Reviewer 2



In this paper, the authors propose a new deep CNN architecture, dubbed Dual Path Networks (DPN). The authors first cast the two of the most popular cnn architectures (ResNet and DenseNet) in the HORNN framework and then propose a simple network augmentation that take the advantages of both architectures. The augmentation is fairly simple. The proposed system achieves good performance results in different vision tasks (classification, detection, semantic segmentation) and the architecture design allows lower complexity and faster training. Although the paper does not have any big novelty (the proposed model is basically a mixture of resnets and densenets), the authors showed different SOA cnn architectures can be cast on the HORNN framework and they achieve SOA results (with improved training) in different vision tasks. The paper contains a bunch of typos and should be re-reviewed in terms of English. I also deeply recommend the authors to release the source code.

Reviewer 3



The authors propose a new network architecture which is a combination of ResNets and DenseNets. They introduce a very informative theoretical formulation which can be used to formulate ResNets, DenseNets and their proposed architecture. The authors present compelling results, analysis and statistics on compelling benchmarks. Pros: (+) The paper is well written with theoretical and empirical results (+) The authors provide useful analysis and statistics (+) The impact of DPNs is shown on a variety of computer vision tasks (+) The performance of the DPNs on the presented vision tasks is compelling Cons: (-) Optional results on MS COCO would make the paper even stronger Network engineering is an important field and it is important that it is done correctly, with analysis and many in depth experiments. The impact of new architectures comes through their generalization capabilities. This paper does a good job on all of the above. Even though there is no groundbreaking novelty in the proposed micro-blocks, the authors provide extensive analysis and statistics for their proposed model. They also present results on a variety of computer vision tasks which makes their paper complete. Especially the analysis of memory usage, training speed and number of flops is very revealing and informative. The authors present results on compelling computer vision tasks, such as ImageNet, Places365 and PASCAL VOC. These are traditionally the most difficult datasets and tasks in computer vision. It would make the paper even stronger if the authors presented results on MS COCO, which is harder than PASCAL VOC.